# Exploring the interplay between colorectal cancer subtypes genomic variants and cellular morphology: A deep-learning approach

Hadar Hezi[1], Daniel Shats[2], Daniel Gurevich[3,4], Yosef E. Maruvka[3,4], Moti Freiman[1]*

1 Faculty of Biomedical Engineering, Technion - Israel Institute of Technology, Haifa, Israel, 2 Faculty of Computer Science, Technion - Israel Institute of Technology, Haifa, Israel, 3 Faculty of Biotechnology and Food Engineering, Technion - Israel Institute of Technology, Haifa, Israel, 4 Lokey Center for Life Science and Engineering, Technion - Israel Institute of Technology, Haifa, Israel

* moti.freiman@technion.ac.il

**Data Availability Statement:** Our source code to replicate the study findings is available at: https://github.com/TechnionComputationalMRILab/MSI_MSS_BP-CNN. TCGA CRC data is available at:

## Abstract

Molecular subtypes of colorectal cancer (CRC) significantly influence treatment decisions. While convolutional neural networks (CNNs) have recently been introduced for automated CRC subtype identification using H&E stained histopathological images, the correlation between CRC subtype genomic variants and their corresponding cellular morphology expressed by their imaging phenotypes is yet to be fully explored. The goal of this study was to determine such correlations by incorporating genomic variants in CNN models for CRC subtype classification from H&E images. We utilized the publicly available TCGA-CRC-DX dataset, which comprises whole slide images from 360 CRC-diagnosed patients (260 for training and 100 for testing). This dataset also provides information on CRC subtype classifications and genomic variations. We trained CNN models for CRC subtype classification that account for potential correlation between genomic variations within CRC subtypes and their corresponding cellular morphology patterns. We assessed the interplay between CRC subtypes' genomic variations and cellular morphology patterns by evaluating the CRC subtype classification accuracy of the different models in a stratified 5-fold cross-validation experimental setup using the area under the ROC curve (AUROC) and average precision (AP) as the performance metrics. The CNN models that account for potential correlation between genomic variations within CRC subtypes and their cellular morphology pattern achieved superior accuracy compared to the baseline CNN classification model that does not account for genomic variations when using either single-nucleotide-polymorphism (SNP) molecular features (AUROC: 0.824±0.02 vs. 0.761±0.04, p<0.05, AP: 0.652±0.06 vs. 0.58±0.08) or CpG-Island methylation phenotype (CIMP) molecular features (AUROC: 0.834±0.01 vs. 0.787±0.03, p<0.05, AP: 0.687±0.02 vs. 0.64±0.05). Combining the CNN models account for variations in CIMP and SNP further improved classification accuracy (AUROC: 0.847 ±0.01 vs. 0.787±0.03, p = 0.01, AP: 0.68±0.02 vs. 0.64±0.05). The improved accuracy of CNN models for CRC subtype classification that account for potential correlation between genomic variations within CRC subtypes and their corresponding cellular morphology as expressed by H&E imaging phenotypes may elucidate the biological cues impacting cancer histopathological imaging phenotypes. Moreover, considering CRC subtypes genomic

https://doi.org/10.5281/zenodo.3832231.
Molecular feature analysis information is available at: https://www.cbioportal.org/.

**Funding:** M.F. acknowledges funding from the Israel Innovation Authority (grant number 73249). Y.E.M. acknowledges funding from the Israel Science Foundation (ISF, grant number 2794/21) and from the Israel Cancer Association (ICA, grant number 20210132)." The funders had no role in study design, data collection and analysis, decision to publish, or preparation of the manuscript.

**Competing interests:** The authors have declared that no competing interests exist.

variations has the potential to improve the accuracy of deep-learning models in discerning cancer subtype from histopathological imaging data.

## Introduction

Colorectal cancer (CRC) stands as the second leading cause of cancer-related deaths, resulting in approximately 0.9 million fatalities worldwide each year [1]. CRC is a heterogeneous disease as evident at multiple levels, including genetic, molecular, cellular, and histopathological variations. The heterogeneity of CRC makes disease management complex and diverse. Recognizing and understanding this heterogeneity is crucial for personalized medicine approaches, guiding treatment decisions, and developing new therapeutic strategies. Specifically, molecular subtyping of CRC into microsatellite instability (MSI) and microsatellite stability (MSS) subtypes is critical in selecting an appropriate immunotherapy protocol to achieve the best treatment response [2, 3].

The gold standard for CRC subtyping into MSI and MSS is DNA sequencing using a Polymerase Chain Reaction (PCR) test [4]. However, this test is expensive, time-consuming, and has limited availability to patients.

Recently, convolutional neural networks (CNNs) [5, 6] have been proposed for automatically determining CRC subtypes from common Hematoxylin and Eosin (H&E) stained histopathological images [7–13]. By leveraging images already produced in regular clinical practices, this method holds promise as a cost-effective and precise solution for identifying CRC subtypes within the existing clinical framework.

However, CNN models to date have primarily focused on CRC subtype classification such as MSI or MSS, essentially assuming a strong correlation between CRC subtypes of MSI and MSS and their histopathological imaging phenotype. Yet, within subtype genomic heterogeneity might be associated with variations in cellular morphology as expressed in the imaging phenotypes. For instance, Zheng et al. [14] proposed that DNA methylation patterns might be discernible from whole slide images, given their impact on cellular morphology in several aspects, such as chromatin organization [15] and the determination of cell identity [16]. However, the correlation between CRC subtype genomic variants and their corresponding cellular morphology as expressed imaging phenotypes is yet to be fully explored.

In this study, we aim to leverage CNN-based classification models to investigate the interplay between molecular and morphological levels. Our main hypothesis is that genomic variations within CRC subtypes of MSI and MSS may impact the H&E image phenotype. We examined this hypothesis by developing and evaluating "biologically-primed" CNN classification models that account for the potential correlation between the genomic variations and the imaging phenotype. This is in contrast to previously proposed models for CRC subtype classification which considered only the CRC subtypes as potential classes, ignoring the heterogeneity within each subtype. To better reflect this, we term our model "biologically-primed," as it integrates biological variations within subtypes, leading to a more comprehensive and precise understanding of CRC subtypes.

In this study, we particularly focused on single-nucleotide-polymorphism (SNP) and CpG-Island methylation phenotype (CIMP) within the MSI and MSS CRC subtypes because of their significant heterogeneity observed within the MSI subtype as indicated by Liu et al. [17]. We then compared the performance of these models to a baseline CNN classification model that does not account for potential correlation between the gnomic variations and the imaging phenotype using the publicly available TCGA-CRC-DX dataset [18] with a stratified 5-fold cross-validation experimental setup.

Our experiments indicate that accounting for potential correlation between genomic variations within CRC subtypes and their imaging phenotype improved CRC subtypes classification accuracy compared to the baseline CNN classification model that does not account for such correlations when considering either SNP or CIMP. These results suggest a correlation between genomic variations within CRC subtypes of MSI and MSS and the tumor morphology and microenvironment as depicted by the H&E images. Further, accounting for genomic variations within CRC subtypes has also the potential to improve the accuracy of deep-learning-based methods for cancer subtypes classification.

It is important to highlight that our model does not directly use genomic data as input. Instead, we represent the CRC subtype class as two distinct classes based on their genomic variations. The genomic variation information is utilized during model training to label the MSI patches as either $MSI_1$ or $MSI_2$. Therefore, while our training phase incorporates molecular subtype information, the inference process depends exclusively on the H&E images, with no additional data used.

The main contributions of the paper are summarized as follows:

- Revealing the correlation between CRC subtypes genomic variations such as SNP and CIMP and cellular morphology expressed by H&E imaging phenotype.

- Introducing CNN models for CRC subtype classification that consider the potential correlation between CRC subtype genomic heterogeneity and cellular morphology as expressed by H&E imaging phenotype.

- Improved accuracy for CNN-based CRC subtype classification

## Related work

During the past few years, a plethora of CNN-based methods were proposed for CRC subtype classification from H&E stained images. Kather et al. [7] were the first to infer CRC molecular sub-types MSI and MSS from H&E images. They divided the images into small patches, performed patch-level classification with CNN, and aggregated the classification results to cope with the giga-pixel size of the images. Their approach achieved moderate success on the TCGA-CRC-DX [18] database (Area under the Receiver operating characteristic (AUROC) per patient of 0.77, n = 360, 18% MSI). Echle et al. [19], used a similar method but further improved the overall classification accuracy by increasing the dataset size through the combination of several databases.

Multiple Instance Learning approaches (MIL) were also proposed to tackle the giga-pixel size of the CNNs. These approaches aim to extract meaningful patches representing the whole slide H&E image. For instance, Bilal et al. [13] employed an iterative 'draw and rank' technique to exclude less informative patches in CRC subtype classification, integrating patch selection during the preprocessing phase. Zhang et al. [20] enhanced classification by clustering patches into various bags followed by bag distillation. Lin et al. [21] introduced interventional bag learning for deconfounded bag-level predictions. Liang et al. [12] integrated spatial locations of adjacent instances for each patch, aiming to minimize both false negatives and positives through an inter-patch messaging mechanism. In the specific context of CRC subtype classification, Lou et al. [11] unveiled a parameter partial sharing network (PPsNet) that merges tumor patch detection with subtype classification, and Schirris et al. [22] combined contrastive self-supervised learning for feature extraction with a variability-aware deep multiple instance learning for classification. For a comprehensive analysis of recent techniques utilizing deep learning for the classification of CRC subtypes from standard H&E stained histopathological images, we refer to the study by Kuntz et al. [8].

Yet, until now, CNN models have predominantly targeted the classification of CRC subtypes, like MSI or MSS, implicitly suggesting a marked correlation between MSI and MSS CRC subtypes and their histopathological imaging characteristics. However, intrinsic genomic variations within these subtypes such as significant heterogeneity in SNP rates and CIMP types observed within the MSI subtype [17] could influence cellular morphology as expressed in their imaging phenotypes. Therefore, this study aims to determine potential correlations between genomic variants of CRC subtypes and cellular morphology as expressed by their H&E imaging phenotype by assessing the benefits of accounting for such potential correlations in CNN models for CRC subtype identification from H&E images.

## Materials and methods

### Data and pre-processing

We utilized the TCGA-CRC-DX dataset [18] and its genomic analysis for all our experiments. This dataset comprises N = 360 patients diagnosed with Colorectal Cancer (CRC-DX). The samples in the dataset are formalin-fixed paraffin-embedded (FFPE) diagnostic slides, stained with H&E. The dataset includes DNA mutations, RNA expressions, and clinical annotations, alongside the H&E images. The dataset was preprocessed as detailed by Kather et al. [7]. The MSI/MSS labels were assigned as per the criteria detailed by Liu et al. [17], referenced in Supplementary Table 2 of Kather et al. [7]. The genomic information including the SNP rates, CIMP types, and Copy number variation (CNV) values was provided by Liu et al. [17] and Cerami et al. [23].

The distribution of patients is illustrated in Fig 1. Initially, from a cohort of 360 patients, Kather et al. [7] randomly selected 100 patients to form the test set. Image patches were extracted from each H&E image following the procedure detailed in their study. To achieve a balanced training set at the patch level, MSS patches were randomly discarded. The composition of the training set is as follows: 39 MSI patients (comprising 15% of the set), represented by 46,704 patches, and 221 MSS patients, also depicted by 46,704 patches. The test set includes

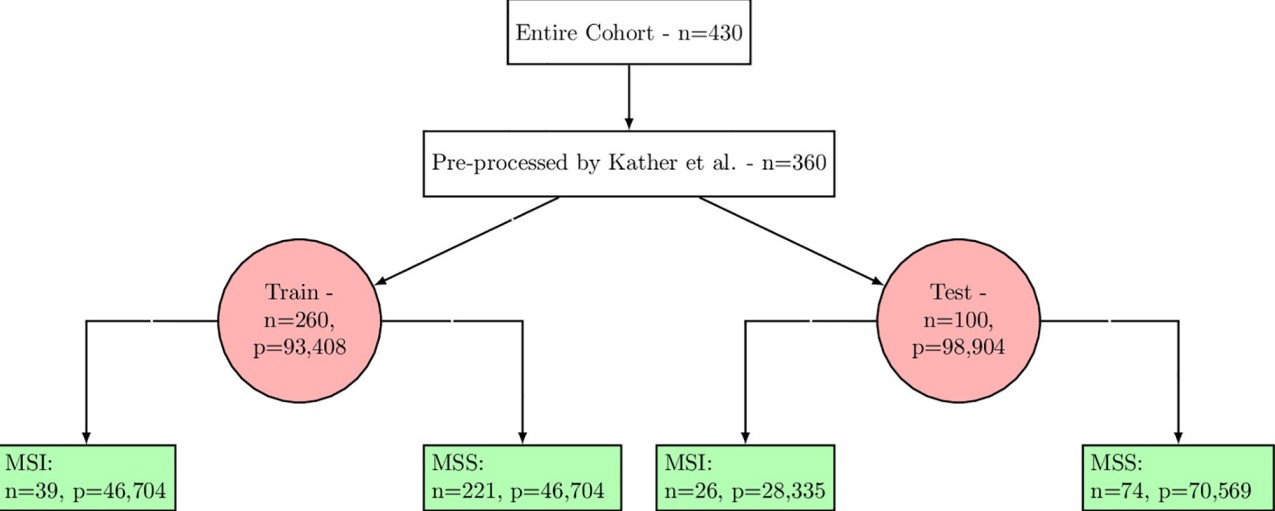

**Fig 1. Summary of the TCGA COAD and READ datasets application: The total cohort encompasses n = 632 patients.** Some patients were excluded due to technical reasons, resulting with n = 430 patients. Out of this, Kather et al. [7] pre-processed and published data for n = 360 patients, segmenting them into a training and a testing set. The training set was balanced at the patch (p) level. For our research, we used stratified cross-validation folds at the patient level. The partitioning into these folds was informed by the novel sub-labels based on SNP rates and CIMP classifications.

26 MSI patients (accounting for 26% of the set) and 28,335 patches, alongside 74 MSS patients symbolized by 70,569 patches.

## CNN architectures for CRC subtype classification

Fig 2 depicts our overall experimental flow. Next, We describe each component in detail.

**Baseline model.** Drawing inspiration from Kather et al. [7], we developed a baseline Convolutional Neural Network (CNN) for patch-level CRC subtype classification into the MSI and MSS classes. Utilizing a transfer learning strategy, we adopted the Inception-v3 network [24] as our primary feature extraction model. Originally trained on the ImageNet dataset, we fine-tuned this model by retraining its last three inception blocks along with the fully connected layers on our dataset.

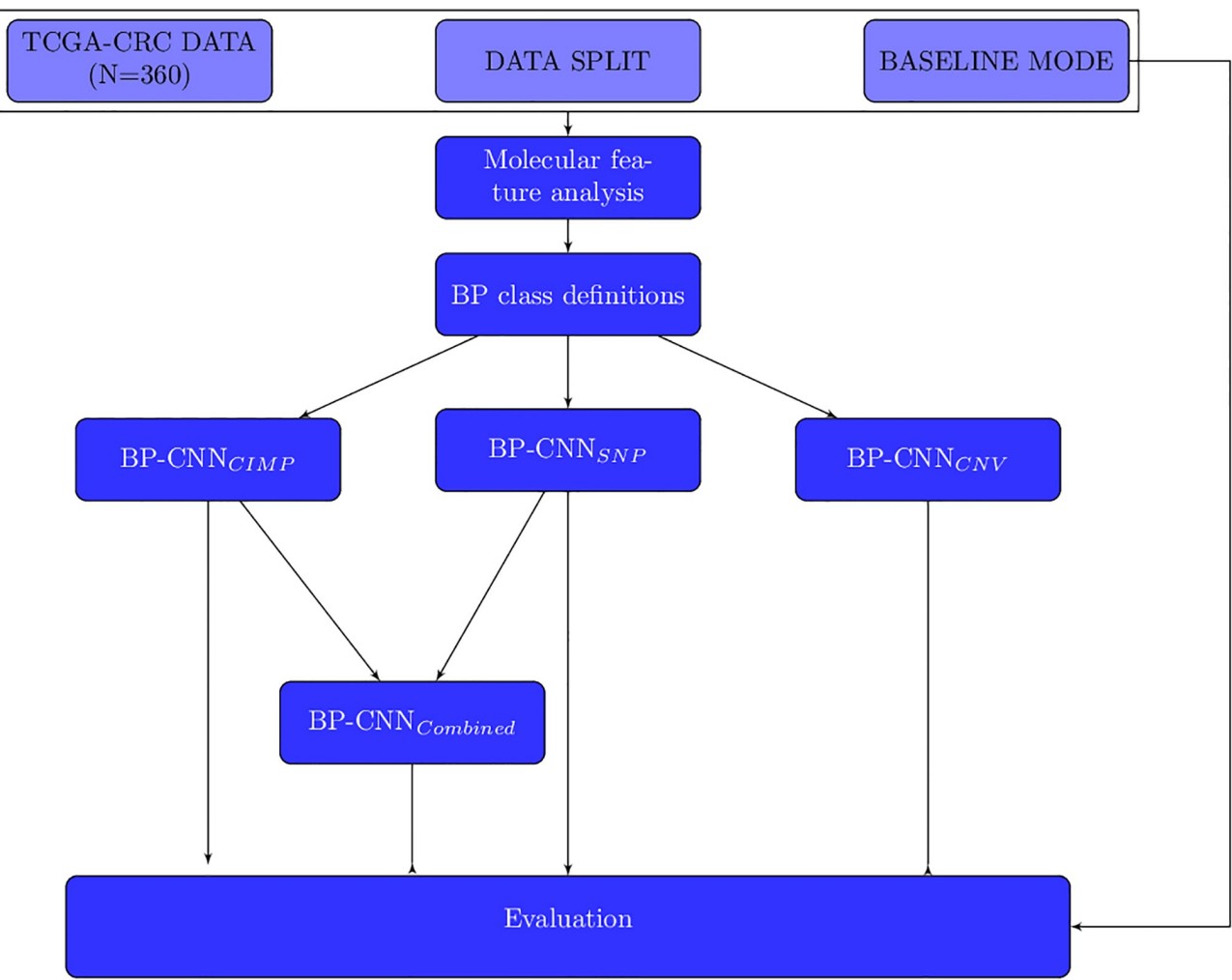

**Fig 2. Experimental flow for our exploration of the interplay between CRC subtypes genomic variants and cellular morphology.** The TCGA-CRC dataset, pre-processed by Kather et al. [7] (N = 360) is split into different sets for analysis. A baseline model is trained, and based on its results, a molecular feature analysis is performed. Based on the analysis we choose to define our data classes based on the ranges and categories of SNP, CIMP and CNV (the BP class definitions step). After the definition, we divide the classes into five stratified folds. Next, three models are trained: BP-CNN_CIMP, BP-CNN_SNP, and BP-CNN_CNV to evaluate the interplay between genomic variations and cellular morphology. The BP-CNN_CIMP, BP-CNN_SNP, and BP-CNN_CNV models classify the data based on CIMP, SNP, and CNV features, respectively. Based on their results, BP-CNN_CIMP and BP-CNN_SNP are further combined into BP-CNN_Combined to incorporate the entire set of genomic variations identified as influencing cellular morphology.

We averaged the classification probabilities of the patches produced by the baseline model to obtain a patient-level classification. Formally, given a classifier $F$, the output for a specific patch $x$ is the probability of being classified as MSI or MSS, denoted as $0 \leq F(x) \leq 1$. For an H&E image $W$ with $N$ extracted patches $((x_1, x_2, \ldots, x_N) \in W)$, we compute the patient-level MSI probability as follows:

$$P_w(MSI) = \frac{\sum_{i=1}^{N} F(x_i)}{N} \tag{1}$$

Fig 3(a) illustrates the architecture of the baseline model used in our research.

**Genomic variations analysis.** We delved into the possible presence of phenotypic variations within classes, theorizing that such diversity might hinder the learning and generalization proficiency of the CNN. Our exploration centered on two dominant attributes: SNPs and the CIMP. Drawing from Liu et al. findings [17], SNP mutations are highly frequent in MSI patients due to their deficiency in the DNA mismatch-repair mechanism. However, MSI samples exhibit significant disparities in SNP density, ranging dramatically from 10 to 17,000, with a median of 1,432. Additionally, the CIMP rate, which influences gene silencing [25], is typically high in MSI patients. Specifically, 60% of MSI patches are categorized as CIMP-High (CIMP-H), while the remaining 40% are non-CIMP-H and categorized as CIMP-low and non-CIMP. This led us to speculate that such intrinsic variances could find reflection in the H&E imaging phenotypes. To maintain a consistent benchmark, we used CNV as a control variable due to its stable nature within MSI tumors.

By charting the biological attribute frequencies from the patch results of our baseline model, our objective was to discern potential characteristic patterns for misclassified patches. We observed that such inaccurately classified patches either aligned closely with the contrasting class or spanned a diverse set of feature values. This observation paved the way for the hypothesis that CNN might have acquired a confined phenotypic spectrum for the classes. To enhance this spectrum for the CNN, we suggested segregating the classes based on these characteristic patterns.

**Biologically-primed models.** Our biologically-primed classification models were developed in the following manner. Instead of a binary classification layer as used in the baseline model, we accounted for potential correlations between genomic variants of CRC subtypes and cellular morphology as expressed by their H&E imaging phenotype by replacing the binary classification layer with a three-class classification layer. This layer identifies one class for MSS and the other two classes for distinct subclasses within the MSI group, based on specific genomic variations. The subclassification generation process is detailed in Alg. 1. Importantly, while molecular subtype details are utilized during training, the inference step remains largely similar to the baseline model. The only difference is the class count. For inference, solely the H&E images are needed, categorizing them into three sets: two MSI subclasses based on the target genomic variation, and MSS. For patient-level classification, we aggregate patch probabilities, considering the higher probability between the MSI subclasses as the MSI probability. Fig 3(b) presents our BP-CNN model.

**Algorithm 1:** Procedure for Generating MSI Sub-class Labels. In this process, $y_i$ represents the original labels provided by the database. $s_i$ refers to the selected feature rates for each patient, as provided by the database. $y_i'$ denotes the newly inferred labels, which are determined based on the feature threshold.

**Algorithm:** Decision of sub-label

```
Input: y₁, ..., yₙ, s₁, ..., sₙ, threshold
Output: y', new labels
for i ← 1 to n do
```

```
if y_i = = MSI then
    if s_i > threshold then
        y'_i ← MSI_2;
    else
        y'_i ← MSI_1;
    end
end
end
```

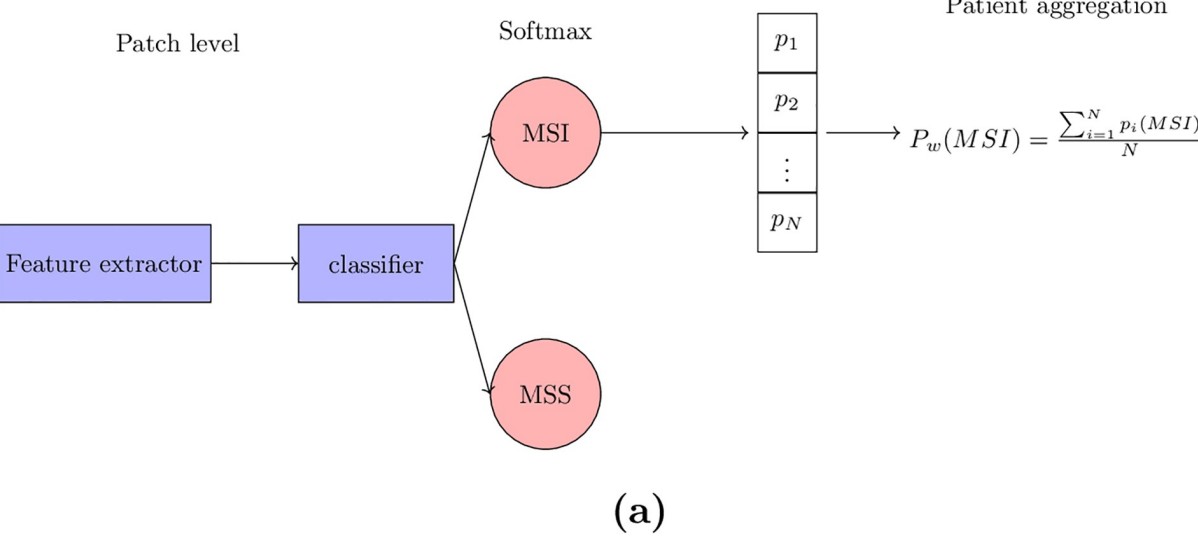

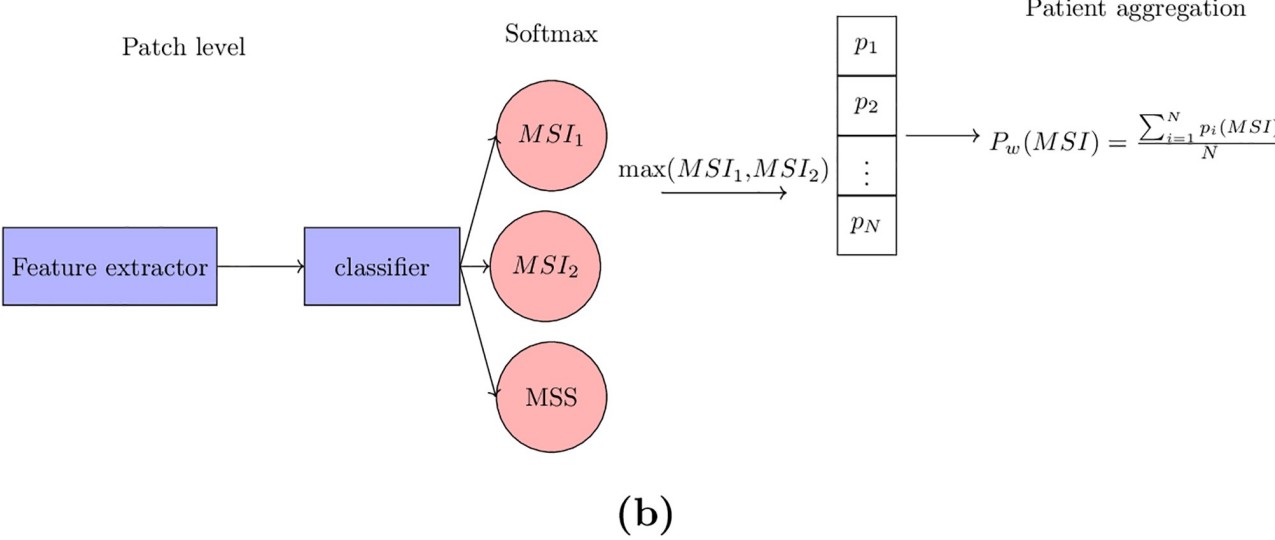

**Fig 3. Model architectures.** (a) Baseline Model Architecture: Patches are input into the Inception-Net [24] for feature extraction, with the last two layers acting as fully connected classifier layers. Outputs are propagated to a softmax layer for determining probabilities. N represents the number of patient patches, while $P_i$ denotes the MSI probability for each patch. The MSI score for each patient, $P_w$, is the average of its corresponding MSI probabilities. (b) Biologically-Primed Model Architecture: Similar to the baseline model, the softmax layer outputs class probabilities at the patch level. However, the MSI probability here is calculated as the maximum value between $MSI_1$ and $MSI_2$ outputs. The calculation of $P_w$ remains the same as in the baseline model.

We engineered three distinct biologically-primed models for our study. The first, BP-CNN$_{SNP}$, partitions MSI patients into two subgroups according to their SNP rate. A range of SNP thresholds from 800 to 1500 were tested on the first fold of the training data to establish an optimal split, and the threshold yielding the highest AUROC on the first training fold's validation dataset was selected. Our second model, BP-CNN$_{CIMP}$, bifurcates the MSI group into CIMP-H and non-CIMP-H subcategories. These subcategories were chosen based on an analysis of patches that were misclassified. Given the low prevalence of CIMP-H patches within the MSS class (5% in the training set and 1% in the test set), we excluded MSS patches exhibiting CIMP-H from the training set. This decision was made with the expectation that it would improve the CNN's ability to recognize the typical MSS phenotype. The BP-CNN$_{CIMP}$ model classifies the patches into three classes: MSS, MSI-CIMP-H, and MSI-NON-CIMP-H.

To confirm that any improvement in classification accuracy was not merely a product of random division, we also implemented a third model, BP-CNN$_{CNV}$. This model segments the MSI group based on each patient's CNV rate, determined through a qualitative evaluation of the CNV distribution.

Lastly, building upon the performance of the individual BP-CNN models exploiting single genomic variation (BP-CNN$_{SNP}$ and BP-CNN$_{CIMP}$), we assembled a combined model that leverages the uncorrelated improvements offered by BP-CNN$_{SNP}$ and BP-CNN$_{CIMP}$. This structure merges the results of the BP-CNN$_{SNP}$ and BP-CNN$_{CIMP}$ through a multi-layer perceptron model (BP-CNN$_{combined}$).

Specifically, the class probabilities were extracted using BP-CNN$_{SNP}$ and BP-CNN$_{CIMP}$. These were used as input to train a multi-layer perceptron (MLP) with a six-dimensional input vector (comprising three class probabilities from each model) for binary classification into MSI or MSS categories. The architecture of this combined model is depicted in Fig 4.

The MLP carries out patch-level classification, while patient-level aggregation is achieved by calculating the average of the respective patch probabilities as described above (Eq 1).

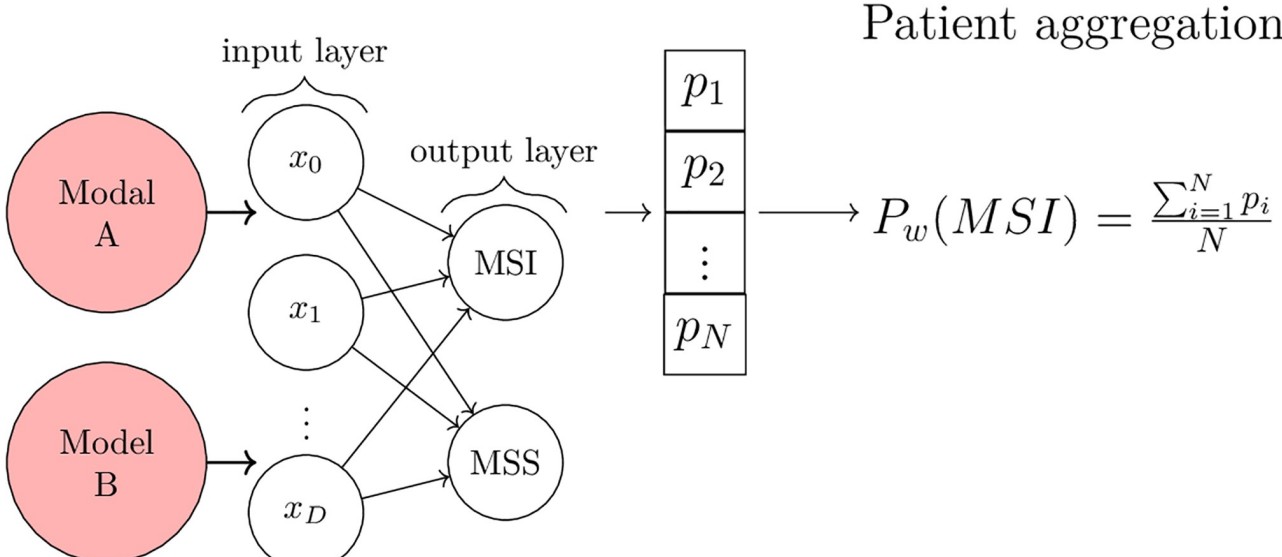

**Fig 4. Our BP-CNN$_{Combined}$ model.** Models A and B represent biologically-primed models informed by two distinct genomic variations. The network outputs from trained and fixed models A and B are concatenated, fed into a linear layer, and then propagated to a softmax layer to yield probabilities. 'N' represents the number of patches for each patient, and $P_i$ indicates the corresponding MSI probabilities for these patches. The MSI score for each patient denoted as $P_w$, is derived from averaging its respective MSI probabilities.

**Training details.** We trained all models using the cross-entropy loss function, employed the Adam optimizer with an initial learning rate of $10^{-4}$, and conducted training over 15 epochs using batches of 64 images. We saved the best model based on validation AUROC. We divided the cross-validation folds using Scikit-learn's stratified folds. To achieve balance at the patch level, we used a random weighted sampler. We implemented the code in PyTorch 1.9 and executed it on Nvidia A100 GPUs using a version 21.04 container image.

## Statistical analysis

We utilized a 5-fold cross-validation experimental setup on the TCGA-CRC-DX training cohort for model development. Given that the distribution of genomic variations is neither consistent nor identical for each type of variation (be it SNP or CIMP), we adopted a stratified k-fold cross-validation method. This ensures a consistent distribution of the CRC subtypes and their internal genomic variations across each fold. Consequently, the composition of the folds varies based on the model under scrutiny. To ensure an equitable comparison, for each experiment (namely, comparing SNP with baseline, CIMP with baseline, and CNV with baseline), we retrained the baseline model using the identical dataset that trained the model of focus. The different models were then tested on the TCGA-CRC-DX test cohort. The performance of the various models in distinguishing between MSI and MSS patients was conducted by utilizing the AUROC metric, along with the average precision (AP) represented as the area beneath the Precision-Recall (PR) Curve and F1-score. We calculated the AUC for each model out of the 5 models developed using the 5-fold cross-validation approach. To determine significant differences in the performance of these models, we applied the Student's paired t-test, setting $p < 0.05$ as the level of significance over the different models.

## Results

### Baseline model

Fig 5 presents our baseline model. It achieved an average AUROC of 0.8 (95% CI, 0.78-0.81) and AP of 0.66 (95% CI, 0.61-0.7) on the 100-patient test set. This outcome is comparable to that reported by Kather et al. [7], who found a median bootstrapped AUROC of 0.77 (95% CI: 0.62–0.87). The slight difference between our model and Kather et al. [7] results could be attributed to several variations between our study and Kather's, including the utilization of a Python implementation instead of Matlab®, patient-level aggregation based on MSI probabilities as proposed by Echle et al [19] rather than the predicted label, and the employment of the Inception v3 model [26] as opposed to Resnet18 [5].

### Genomic variations analysis

The number of SNPs and the CIMP category are extracted from a patient's DNA sample in the TCGA. The range of SNPs among patients varied from 10 to 17000.

Fig 6 presents three molecular features of our CRC patients at the patient level, plotted against the patch classification from our baseline model. In this figure, the MSI class is denoted as the positive class, and the MSS class is denoted as the negative class. The boxplot of the SNP rates concerning the test set's baseline classification results is displayed in Fig 6a. The MSS class consistently shows a low SNP rate, regardless of the model classification outcome. This aligns with prior research indicating that MSS patients usually lack a deficiency in the DNA mismatch repair mechanism [17]. Conversely, for the MSI class, there's a noticeable difference in the SNP distribution between true-positive (TP) and false-negative (FN) classifications. The TP group has a marginally higher median with limited variance, whereas the FN group

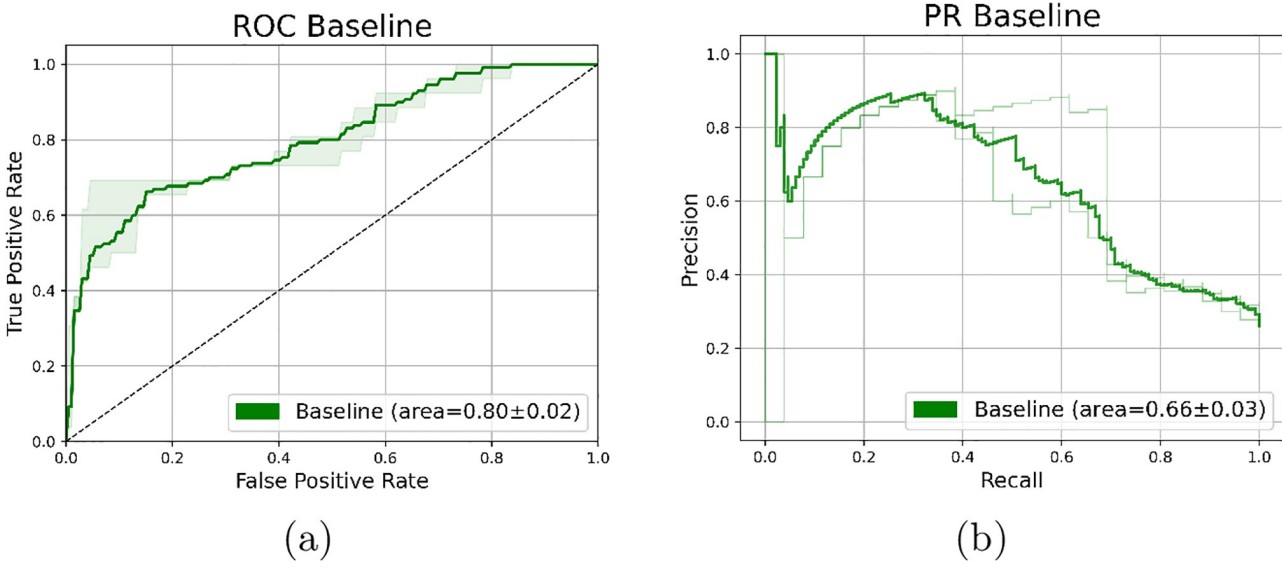

**Fig 5. Baseline model results for per-patient classification of the test set validated over 5-folds.** Average and 95% CI curves: (a) ROC curve, (b) PR curve.

demonstrates a wider range of variation. The threshold of the optimal split for our BP-CNN$_{SNP}$ model was found to be 1200. Fig 6b showcases the distribution of methylation types based on the baseline classification of the test set. Highly methylated (CIMP-H) samples are rare in MSS, present in only 1% of the MSS patches, but are prevalent in MSI, accounting for 59% of the MSI patches. Notably, among the MSI patches that are non-CIMP-H, a substantial portion (76%) was incorrectly classified as MSS (negative class). Therefore we chose to distinguish 2 MSI sub-categories: CIMP-H and non-CIMP-H. Fig 6c presents the CNV distribution in patches as classified by the baseline model. MSS patches display elevated CNV rates with notable variability, in contrast to the MSI patches which exhibit consistently low CNV rates with slight deviations. We chose a threshold of 0.005, determined through a qualitative evaluation of the CNV distribution.

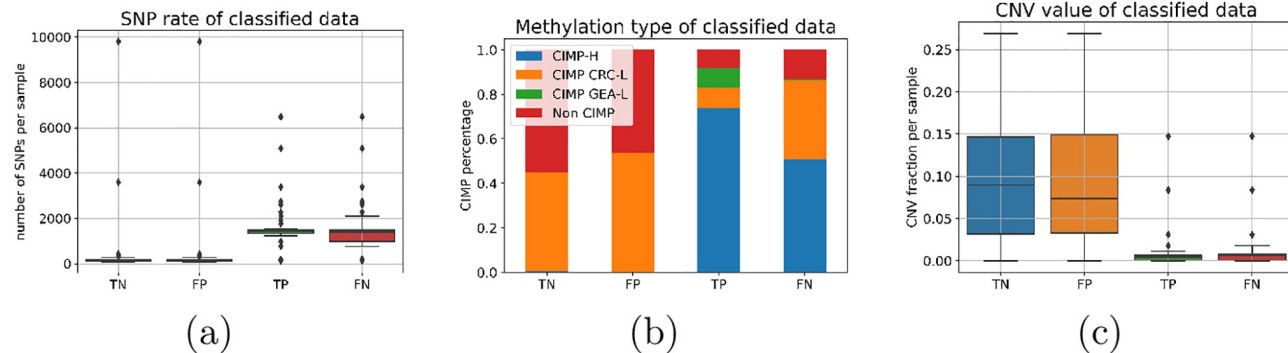

**Fig 6. The distribution of patient-level molecular features in the test set, categorized based on the patch-level classification by the baseline model.** The x-axis indicates the classification of patches, while the y-axis denotes the molecular level determined at the patient level. Here, **MSI** serves as the **positive** class and **MSS** as the **negative** class: (a) A boxplot illustrating SNP rates for each patch. The y-axis quantifies the cumulative count of SNPs throughout the DNA sample. (b) A bar plot depicting the methylation types for each patch. The y-axis showcases the distribution of various methylation types across classification categories. (c) A boxplot highlighting the CNV rates for patches, with the y-axis measuring the proportion of the DNA sample that manifests CNV.

## Biologically-primed models

The BP-CNN$_{SNP}$ model outperformed the baseline model, achieving an AUROC of 0.824 ±0.02 (95% CI 0.79-0.86) compared to 0.761±0.04 (95% CI 0.68-0.8) on a test set of 100 patients across 5 stratified folds sessions. This increase was statistically significant (paired t-test, p<0.05). The AP for the model was 0.652±0.06 (95% CI 0.58-0.72), whereas the baseline's was 0.58±0.08 (95% CI 0.46-0.67). The F1 scores for the model and baseline were 0.65±0.04 and 0.61±0.05, respectively. However, differences in AP and F1 scores did not reach the statistical significance level. Fig 7a presents the ROC and PR curves (average and the 95% CI) for the test set across different 5 stratified fold sessions. Fig 8 depicts the distribution of the AUROC, AP, and F1 on the test set across the different training sessions.

Similarly, the BP-CNN$_{CIMP}$ model achieved a significantly higher AUROC than its corresponding baseline model on the 100 patient test set over the five stratified fold training sessions (0.834±0.01 (95% CI 0.81-0.85) vs. 0.787±0.03 (95% CI 0.72-0.81), paired t-test, p<0.05), a higher AP score (0.687±0.02 (95% CI 0.65-0.72) vs. 0.64±0.05 (95% CI 0.55-0.71)) and a higher F1 score compared to the baseline model (0.71±0.03 vs. 0.63±0.06). Yet, the difference did not reach the pre-defined significant level. Fig 7b showcases the ROC and PR curves (average and 95% CI) for the test set over the various training sessions. Fig 8 depicts the distribution of the AUROC, AP and F1 on the test set across different training sessions.

Conversely, the BP-CNN$_{CNV}$ model lagged behind its corresponding baseline model on the 100-patient test set over five stratified fold training sessions (AUROC: 0.793±0.03 (95% CI 0.75-0.85) vs. 0.801±0.02 (95% CI 0.75-0.83), AP: 0.63±0.09 (95% CI 0.51-0.76) vs. 0.64±0.07 (95% CI 0.51-0.72)). The BP-CNN$_{CNV}$ model's F1 score was slightly higher than its baseline model's (0.64±0.03 vs. 0.62±0.04). Fig 7c presents the ROC and PR curves (average and 95%

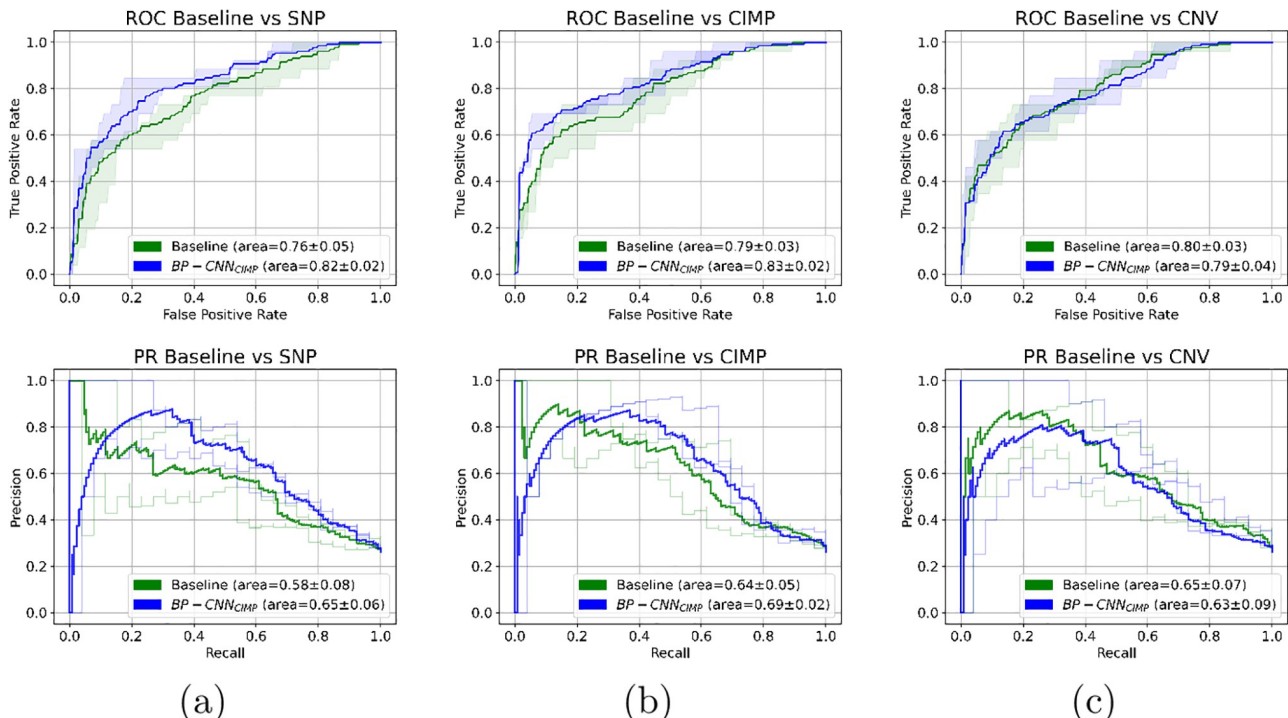

**Fig 7. Average and 95% CI ROC and PR curves for per-patient classification using: (a) the BP-CNN$_{SNP}$ model compared to its corresponding baseline model, (b) the BP-CNN$_{CIMP}$ model compared to its corresponding baseline model, and (c) the BP-CNN$_{CNV}$ model compared to its corresponding baseline model.**

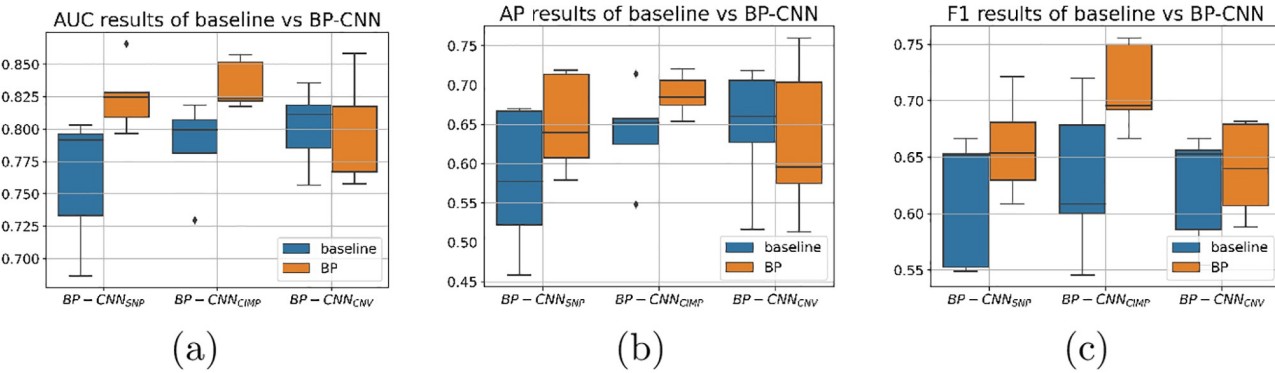

**Fig 8. Box-plot visualization of (a) AUROC results, (b) AP results and (c) F1-scores for per-patient classification, comparing the biologically primed models with their corresponding baseline model on the test set over different training sessions.** It's worth noting that due to the stratified k-fold approach used to partition the training data across sessions, the performance of the baseline model can vary between experiments.

CI) for the test set over various training sessions and Fig 8 presents the AUROC, AP, and F1 distribution on the test set across these sessions. All the differences were not statistically significant.

Fig 9 presents the confusion matrices for per-patient classification on the test set, averaged over the training sessions, for the BP-CNN$_{SNP}$, the BP-CNN$_{CIMP}$ and their corresponding baseline models. The BP-CNN$_{SNP}$ model was more proficient at classifying MSI patients, while the BP-CNN$_{CIMP}$ model excelled at classifying MSS patients.

## BP-CNN$_{Combined}$ model

The BP-CNN$_{combined}$ outperformed the baseline model significantly on the 100 patient test set over five stratified fold training sessions, achieving an AUROC of 0.847±0.01 (95% CI 0.82-0.87) versus 0.787±0.03 (95% CI 0.72-0.81), as shown by a paired t-test with p<0.01. The AP score was 0.68±0.02 (95% CI 0.63-0.71) versus vs. 0.64±0.05 (95% CI 0.55-0.71). The average F1 score on the test set, across the different training sessions for the BP-CNN$_{combined}$, was 0.71 ±0.02, compared to 0.63±0.06 for the baseline model. Yet, the difference of the AP nor the F1 did not reach the pre-defined significance level. Fig 10 displays the ROC and PR curves and the boxplots of the AUROC, AP and F1 scores of the test set over the various training sessions.

Fig 11 presents the histograms of the patch-level probabilities of selected patients, misclassified by the baseline model but accurately classified by our proposed models. The patches' distributions produced by our proposed models were more aligned with the patient-level reference classification compared to the distributions computed by the baseline models.

Fig 12 illustrates patches from patients that were misclassified by the baseline model but accurately classified by our BP-CNN$_{Combined}$ model (top row) and those misclassified by both models (bottom row). Patches (a, b), correctly identified as MSI by our BP-CNN$_{Combined}$ model but mislabeled as MSS by the baseline, display pronounced nuclear pleomorphism and prominent nucleoli—hallmarks of MSI. This suggests that the BP-CNN$_{Combined}$ model is more adept at detecting these subtle yet crucial variations than the baseline model. Conversely, patches (c, d) that were correctly identified as MSS by our BP-CNN$_{Combined}$ model but misclassified as MSI by the baseline likely exhibit characteristics such as poor gland formation and high intra-tumoral lymphocytes, which are typically indicative of MSI, demonstrating that our BP-CNN$_{Combined}$ model can correctly interpret these complex features, where the baseline model fails.

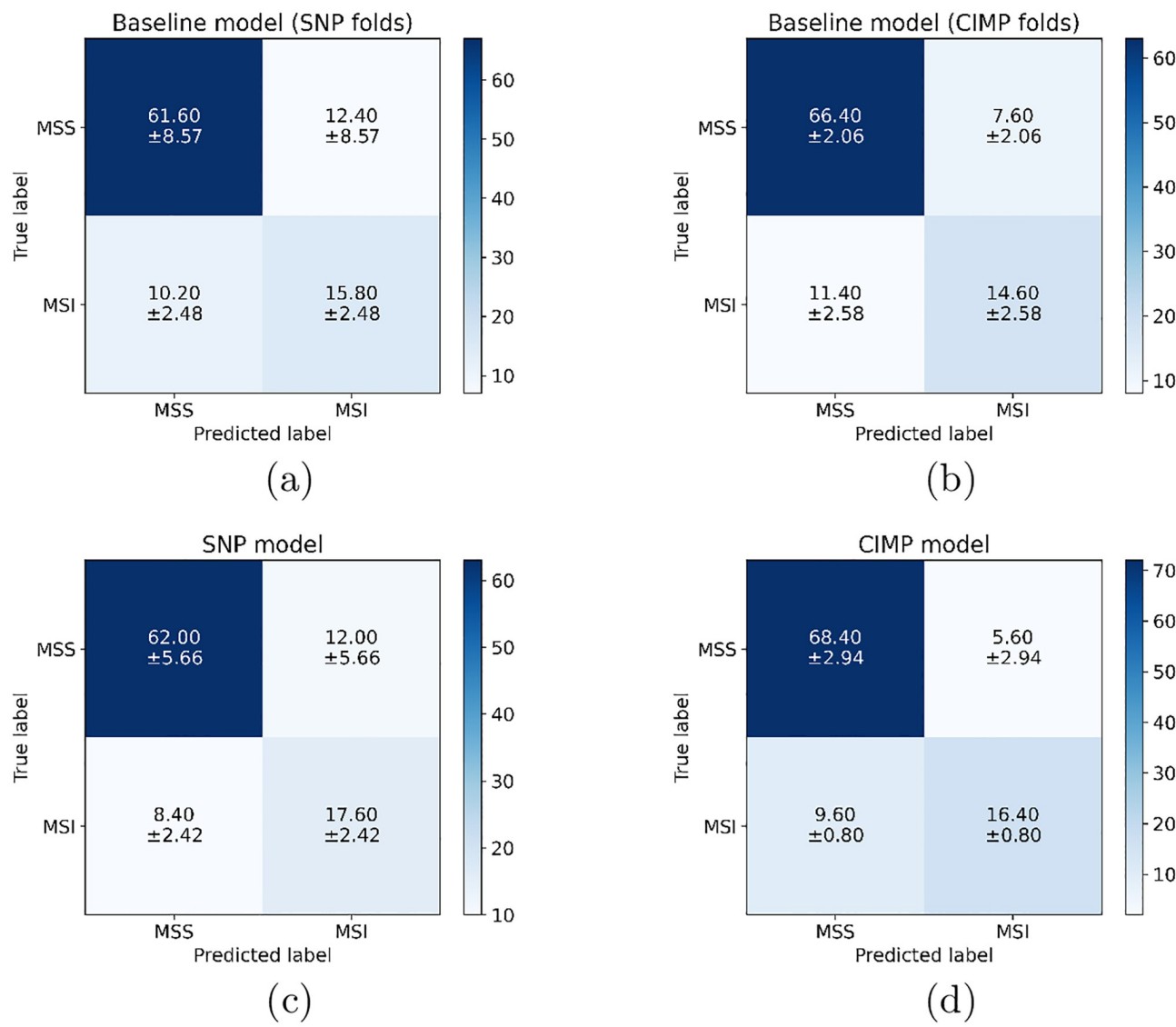

**Fig 9. Confusion matrices of the patient-level predictions for the different models.** Each matrix represents an average from the test set over various training sessions. The threshold for MSI prediction is determined by the best F1 score over the folds. (a) Baseline model corresponding to the BP-CNN$_{SNP}$ folds. (b) Baseline model corresponding to the to BP-CNN$_{CIMP}$ folds. (c) BP-CNN$_{SNP}$ model. (d) BP-CNN$_{CIMP}$ model.

The patches misclassified by both models (bottom row) likely exhibit mixed features or subtle signs that pose classification challenges, including moderate gland formation with irregularities and characteristics that straddle the line between MSI and MSS, such as moderate nuclear pleomorphism and visible but subdued nucleoli. These ambiguous features contribute to the confusion in model classifications.

## Discussion

Differentiating between CRC subtypes using H&E stained histopathological image analysis is paramount for the cost-effective, widespread implementation of personalized treatment plans for patients [2]. Recently, the employment of CNN-based techniques has emerged as an automated method for classifying H&E stained histopathological images of CRC [7–13]. Thus far,

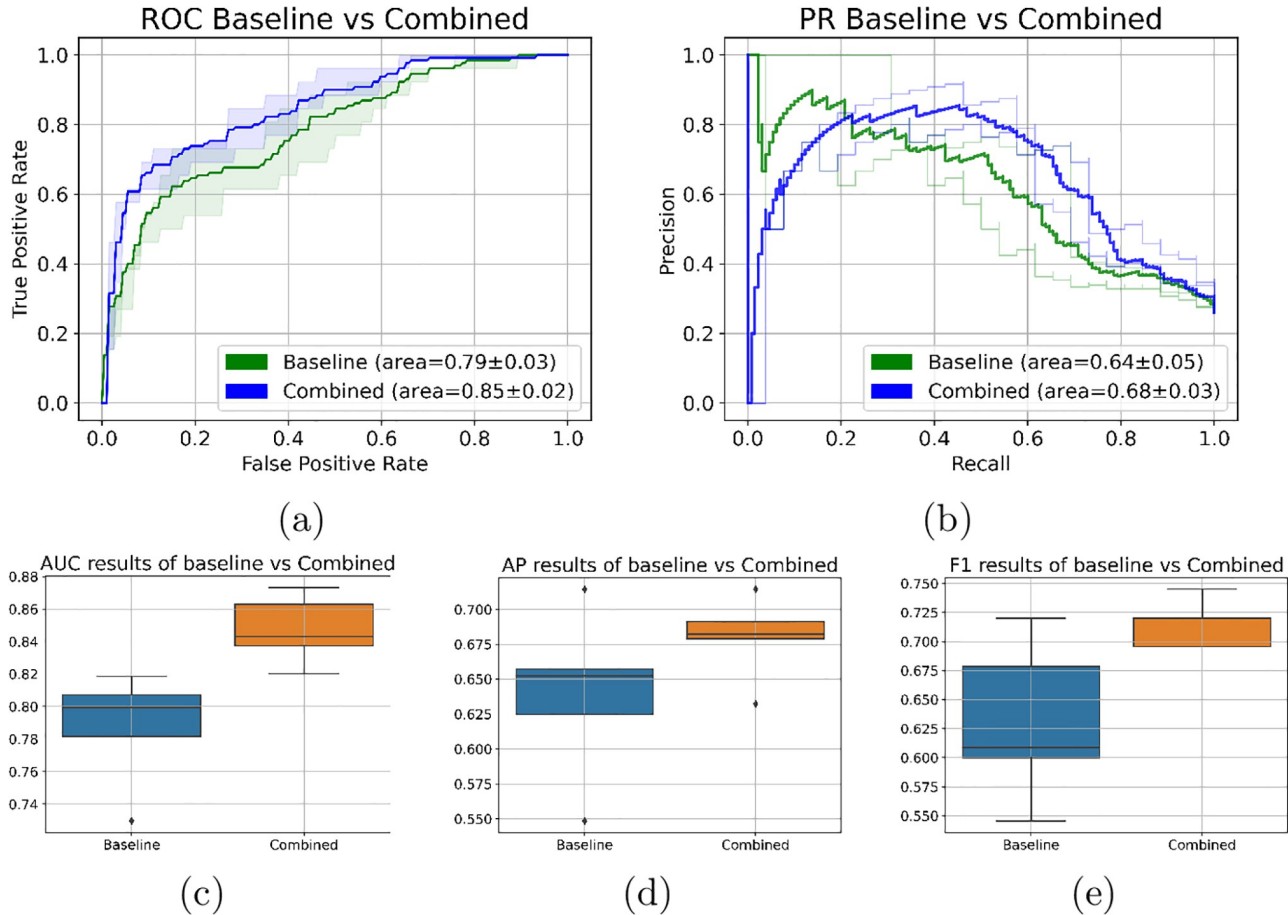

**Fig 10. Average and 95% CI ROC and PR curves for per-patient classification using the BP-CNN<sub>Combined</sub> model compared to the baseline model.**
(a) ROC curve. (b) PR curve. (c), (d) and (e) are the 5-fold results comparison of the AUROC, AP, and F1 results respectively.

CNN models have mainly concentrated on classifying CRC subtypes like MSI or MSS, presuming a robust correlation between MSI and MSS CRC subtypes and their histopathological imaging characteristics. However, genomic heterogeneity within subtypes could correlate with differences in cellular morphology as expressed in the H&E imaging phenotypes.

The present study reveals the correlation between the SNP and CIMP genomic variants of CRC subtypes and their cellular morphology as expressed by their H&E imaging phenotype. Our experimental results show a significant enhancement in the AUC results for differentiating CRC into MSI and MSS subtypes when utilizing our BP-CNN approach along with CNNs compared to baseline models. This enhancement suggests that the SNP and CIMP genomic variations influence the tumor cellular morphology as expressed in the H&E stained histopathological imaging phenotype, whereas the CNV does not. A fascinating observation is that the inclusion of the SNP molecular feature in the CNN bolsters the classification of MSI patients, while the addition of the CIMP molecular feature improves the classification of MSS patients. This suggests that the integration of both SNP and CIMP into a unified model could lead to superior overall accuracy in classifying CRC subtypes.

However, the direct inclusion of multiple genomic variations in the BP-CNN approach can be challenging due to the overlap of patients across classes. We therefore merged the classification outcomes of the SNP and CIMP-based models using a feed-forward multi-layer

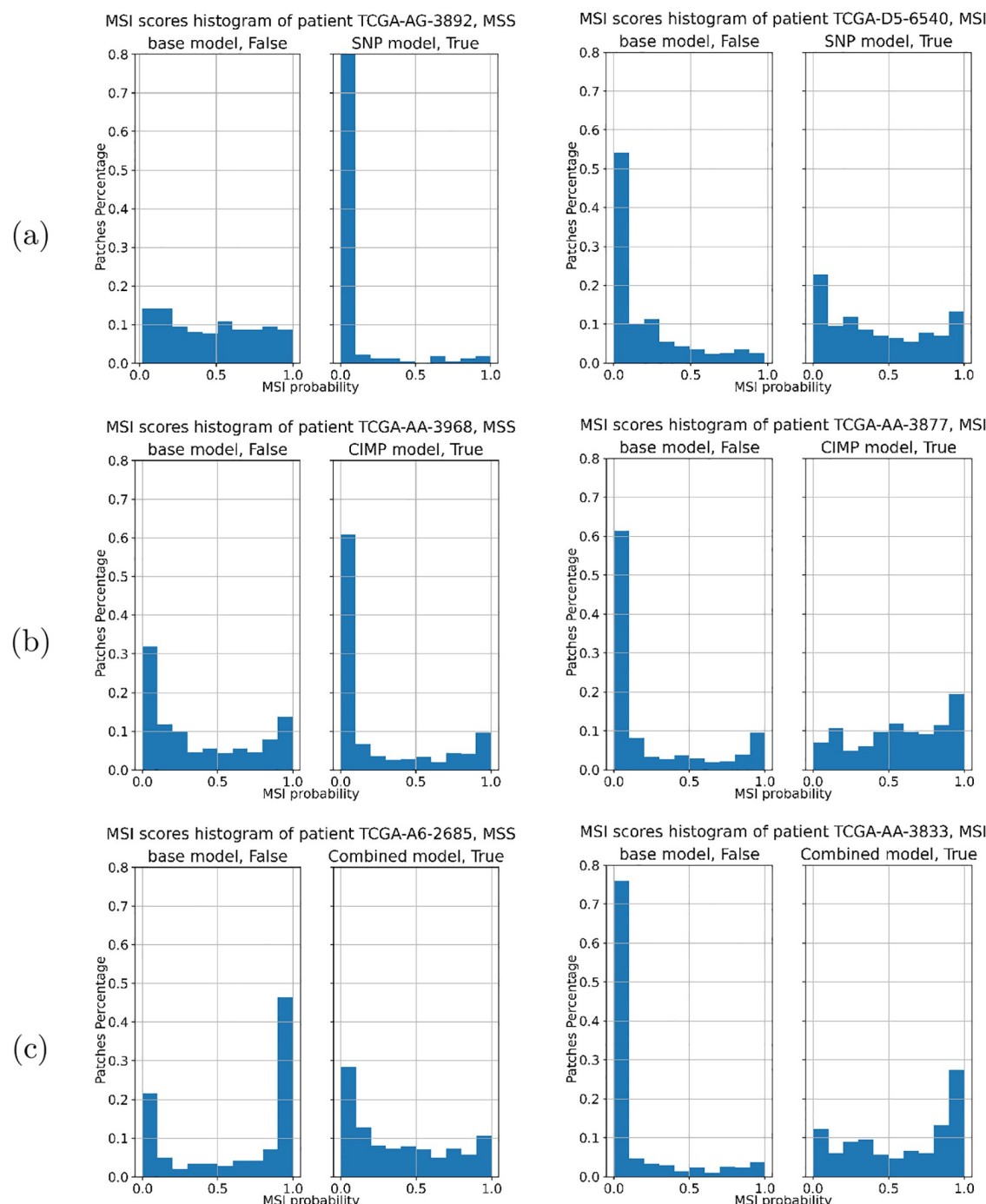

**Fig 11. A histogram showcasing the MSI scores for patches from selected patients, misclassified by the baseline model but accurately classified by our proposed models.** The x-axis represents the patch MSI probabilities given by the CNN, while the y-axis denotes the count of patches, normalized to the total number of patches for each patient. The comparisons are between (a) the Baseline and BP-CNN$_{SNP}$ model, (b) the Baseline and BP-CNN$_{CIMP}$ model, and (c) the Baseline and BP-CNN$_{Combined}$ model.

perceptron model. Our experiments corroborate that the combined model excels over the baseline model in the accurate classification of CRC subtypes. It's worth mentioning that although there are various methods to aggregate the results from the SNP and CIMP-based models, our primary concern was to investigate the link between genomic variations and the

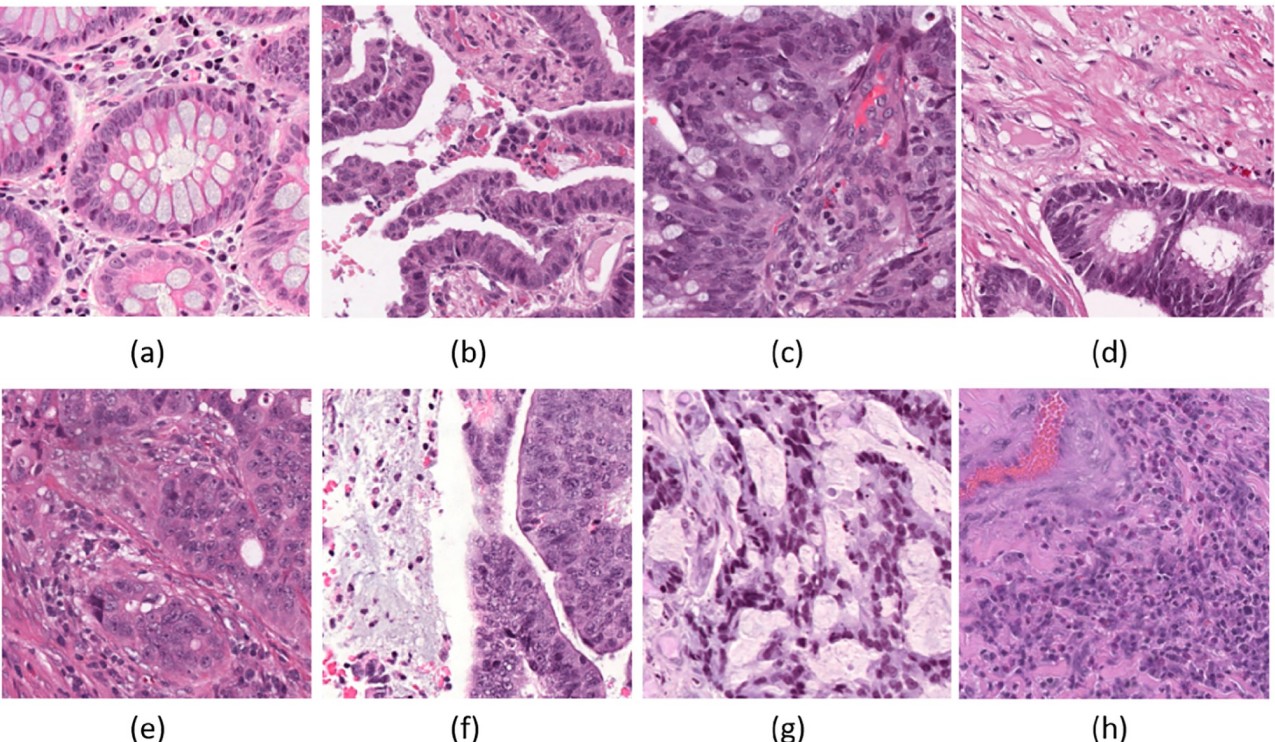

**Fig 12. Patches of patients that were miss-classified by our models.** Top row: patches of patients that were misclassified by the Baseline model and correctly classified by the BP-CNN$_{Combined}$ model. (a) TCGA-AA-3833, Baseline: MSS, BP-CNN$_{Combined}$: MSI, reference: MSI (SNP<1200), (b) TCGA-AY-6197, Baseline: MSS, BP-CNN$_{Combined}$: MSI, reference: MSI (CIMP-low), (c) TCGA-A6-2685, Baseline: MSI, BP-CNN$_{Combined}$: MSS, reference: MSS, (d) TCGA-NH-A6GC, Baseline: MSI, BP-CNN$_{Combined}$: MSS, reference: MSS. Bottom row: patches of patients that were misclassified by both the Baseline model and the BP-CNN$_{Combined}$ model. (e) TCGA-A6-2686, Baseline: MSS, BP-CNN$_{Combined}$: MSS, reference: MSI, (f) TCGA-AG-A02N, Baseline: MSS, BP-CNN$_{Combined}$: MSS, reference: MSI, (g) TCGA-AG-3881, Baseline: MSI, BP-CNN$_{Combined}$: MSI, reference: MSS, (h) TCGA-DC-6682, Baseline: MSI, BP-CNN$_{Combined}$: MSI, reference: MSS.

imaging phenotype. Therefore, the precise technique of model integration is secondary to our main objective, rather than aiming for the optimal CRC subtype classification.

It is also vital to emphasize that although our training phase incorporates molecular subtype information, the inference process depends solely on H&E images, without the need for any additional data. Hence, our methodology is in alignment with prior methods [7, 13] when it comes to predicting MSI/MSS status from H&E images alone.

This study is subject to several limitations. Firstly, we relied exclusively on the CRC TCGA dataset as pre-processed by Kather et al. [7]. Consequently, the extrapolation of these findings to other datasets should be approached with caution. Additionally, the use of different pre-processing methodologies could also impact the study outcomes.

A further limitation is our examination of a limited number of molecular features. Drawing inspiration from Liu et al. work [17], we focused on two specific features, SNP and CIMP, as potential factors influencing the appearance of H&E stained histopathological images of CRC. CNV was also included as a control feature. It would be advantageous to explore the impact of additional molecular features on the phenotype of H&E stained histopathological images of CRC.

Moreover, factors such as age, gender, and tumor location might also influence the appearance of H&E images. Integrating these aspects could enhance the ability of CNN-based models to effectively classify CRC subtypes using these images.

Finally, we showcased the interaction between genomic variations in CRC subtypes and the H&E imaging phenotype by gauging the enhancement in CRC subtype classification accuracy using rudimentary CNNs [7]. However, these CNN methods aggregate patch-level classifications rather than employing advanced MIL techniques [11–13, 20–22]. While using a simple CNN approach increases the robustness of our findings, an intriguing avenue for future research would be to investigate the added-value of leveraging this interplay in improving MIL models for CRC subtype classification.

## Conclusion

Our study highlighted the influence of SNP and CIMP variations on the tumor cellular morphology as expressed by their H&E images by evaluating the classification accuracy of Biologically Primed Convolutional Neural Networks (BP-CNN) that considers the potential impact of genomic variations on the appearance of H&E stained histopathological images of colorectal cancer (CRC) in comparison to baseline CNN classification models. The results and approach of this study could be invaluable to researchers investigating the connection between genetic mutations and image characteristics in various types of cancer. Furthermore, these findings can be leveraged by engineers striving to enhance the accuracy of CNN-based methods for classifying cancer subtypes using H&E stained histopathological images.

## Author Contributions

**Conceptualization:** Yosef E. Maruvka, Moti Freiman.

**Data curation:** Hadar Hezi, Daniel Shats, Daniel Gurevich.

**Formal analysis:** Hadar Hezi, Daniel Gurevich, Yosef E. Maruvka, Moti Freiman.

**Funding acquisition:** Yosef E. Maruvka, Moti Freiman.

**Investigation:** Hadar Hezi, Daniel Gurevich.

**Methodology:** Hadar Hezi, Yosef E. Maruvka, Moti Freiman.

**Software:** Hadar Hezi, Daniel Shats, Daniel Gurevich.

**Supervision:** Moti Freiman.

**Writing – original draft:** Hadar Hezi, Moti Freiman.

**Writing – review & editing:** Daniel Gurevich, Yosef E. Maruvka, Moti Freiman.

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
