## [Decision Letter · Decision Letter 0]

7 May 2024

PONE-D-24-07948Exploring the Interplay Between Colorectal Cancer Subtypes Genomic Variants and Cellular Morphology: A Deep-Learning ApproachPLOS ONE

Dear Dr. Freiman,

Thank you for submitting your manuscript to PLOS ONE. After careful consideration, we feel that it has merit but does not fully meet PLOS ONE’s publication criteria as it currently stands. Therefore, we invite you to submit a revised version of the manuscript that addresses the points raised during the review process.

We look forward to receiving your revised manuscript.

Kind regards,

Tao Huang

Academic Editor

PLOS ONE

Journal Requirements:

".F. acknowledges funding from the Israel Innovation Authority (grant number 73249) and from Microsoft Education and the Israel Inter-university computation center (IUCC). 

Y.E.M. acknowledges funding from the Israel science foundation (ISF, grant number 2794/21) and from the Israel cancer association (ICA, grant number 20210132)."

"M.F. acknowledges funding from the Israel Innovation Authority (grant number 73249) and from Microsoft Education and the Israel Inter-university computation center (IUCC). Y.E.M. acknowledges funding from the Israel science foundation (ISF, grant number 2794/21) and from the Israel cancer association (ICA, grant number 20210132)."

".F. acknowledges funding from the Israel Innovation Authority (grant number 73249) and from Microsoft Education and the Israel Inter-university computation center (IUCC). 

Y.E.M. acknowledges funding from the Israel science foundation (ISF, grant number 2794/21) and from the Israel cancer association (ICA, grant number 20210132)."

Reviewers' comments:

Reviewer's Responses to Questions

**Comments to the Author**

1. Is the manuscript technically sound, and do the data support the conclusions?

Reviewer #1: Partly

2. Has the statistical analysis been performed appropriately and rigorously? 

Reviewer #1: Yes

3. Have the authors made all data underlying the findings in their manuscript fully available?

Reviewer #1: Yes

4. Is the manuscript presented in an intelligible fashion and written in standard English?

Reviewer #1: Yes

5. Review Comments to the Author

Reviewer #1: This manuscript described a development of artificial intelligence (AI) for classification of subtype of colorectal cancer (CRC) using tumor tissue image of H&E stains. By incorporating genomic information for classification, the AI models performed higher accuracy than baseline AI models. This is novel and valuable. Authors showed the results based on statistical data. However, no example images were shown to account for the success of classification. The description would help scientists to recognize what structures in CRC image of microsatellite instability are important and are recognized by the AI models.

Major point

Authors would need to add figures on example of images which successfully recognized by the new AI model. Those images were failed to be classified appropriately by baseline AI models. Also, authors would need to show an example image which were not classified appropriately even if new AI models were conducted for proper classification. Comments on the cellular morphology would be interesting for readers.

Minor point

1. Line 240. Please provide the full name of abbreviation CNV.

6. PLOS authors have the option to publish the peer review history of their article (what does this mean?). If published, this will include your full peer review and any attached files.

Reviewer #1: No

---

## [Author Response · Author response to Decision Letter 0]

22 May 2024

Reviewer #1 comments:

1. "This manuscript described a development of artificial intelligence (AI) for classification of subtype of colorectal cancer (CRC) using tumor tissue image of H\\&E stains. By incorporating genomic information for classification, the AI models performed higher accuracy than baseline AI models. This is novel and valuable. Authors showed the results based on statistical data.''

Response: We thank the reviewer for the positive feedback on our manuscript. 

2. "However, no example images were shown to account for the success of classification. The description would help scientists to recognize what structures in CRC image of microsatellite instability are important and are recognized by the AI models."

Response: We thank the reviewer for this comment. In response, we have added a new figure to the revised manuscript (Fig. 11) that illustrates examples of patches misclassified by the baseline model but correctly identified by our proposed model, along with patches that neither model classified accurately. The new figure has been included below for the convenience of the reviewer.

``Figure 11 depicts patches from patients that were classified incorrectly by the baseline model but correctly by our BP-CNN\\textsubscript{Combined} model (top row) as well as patches from patients that were classified incorrectly by both models (bottom row).''

3. "Authors would need to add figures on example of images which successfully recognized by the new AI model. Those images were failed to be classified appropriately by baseline AI models."

Response: Thank you for your feedback. As noted earlier, we have included the requested figure in the revised version of the manuscript.

4. "Comments on the cellular morphology would be interesting for readers."

Response: We thank the reviewer for this comment. We have expanded our discussion on cellular morphology within the correctly and incorrectly classified patches in the results section of our manuscript. Specifically, we have added the following paragraphs:

``Patches (a, b), correctly identified as MSI by our BP-CNN\\textsubscript{Combined} model but mislabeled as MSS by the baseline, display pronounced nuclear pleomorphism and prominent nucleoli—hallmarks of MSI. This suggests that the BP-CNN\\textsubscript{Combined} model is more adept at detecting these subtle yet crucial variations than the baseline model. Conversely, patches (c, d) that were correctly identified as MSS by our BP-CNN\\textsubscript{Combined} model but misclassified as MSI by the baseline likely exhibit characteristics such as poor gland formation and high intra-tumoral lymphocytes, which are typically indicative of MSI, demonstrating that our BP-CNN\\textsubscript{Combined} model can correctly interpret these complex features, where the baseline model fails.

The patches misclassified by both models (bottom row) likely exhibit mixed features or subtle signs that pose classification challenges, including moderate gland formation with irregularities and characteristics that straddle the line between MSI and MSS, such as moderate nuclear pleomorphism and visible but subdued nucleoli. These ambiguous features contribute to the confusion in model classifications.''

5. "Line 240. Please provide the full name of abbreviation CNV."

Response: We thank the reviewer for indicating this. We provided the full name of CNV on the first appearance (line 107 in the original submission) in the revised version. the updated line is: ``The genomic information including the SNP rates, CIMP types, and Copy number variation (CNV) values was provided by Liu et al.''

---

## [Decision Letter · Decision Letter 1]

24 Jun 2024

PONE-D-24-07948R1Exploring the Interplay Between Colorectal Cancer Subtypes Genomic Variants and Cellular Morphology: A Deep-Learning ApproachPLOS ONE

Dear Dr. Freiman,

Thank you for submitting your manuscript to PLOS ONE. After careful consideration, we feel that it has merit but does not fully meet PLOS ONE’s publication criteria as it currently stands. Therefore, we invite you to submit a revised version of the manuscript that addresses the points raised during the review process.

We look forward to receiving your revised manuscript.

Kind regards,

Tao Huang

Academic Editor

PLOS ONE

Journal Requirements:

Reviewers' comments:

Reviewer's Responses to Questions

**Comments to the Author**

1. If the authors have adequately addressed your comments raised in a previous round of review and you feel that this manuscript is now acceptable for publication, you may indicate that here to bypass the “Comments to the Author” section, enter your conflict of interest statement in the “Confidential to Editor” section, and submit your "Accept" recommendation.

Reviewer #1: All comments have been addressed

2. Is the manuscript technically sound, and do the data support the conclusions?

Reviewer #1: Yes

3. Has the statistical analysis been performed appropriately and rigorously? 

Reviewer #1: Yes

4. Have the authors made all data underlying the findings in their manuscript fully available?

Reviewer #1: Yes

5. Is the manuscript presented in an intelligible fashion and written in standard English?

Reviewer #1: Yes

6. Review Comments to the Author

Reviewer #1: The revised manuscript responded reviewer’s comments precisely and appropriately for the better description with additional figures. Thereby, the great value of new AI model was visible and recognized by readers. The new approach of making AI model would be interested in the community.

As small points, I would make additional comments for readers.

Minor points

1. Please add a description on what is referred by the positive in Fig 4. MSI? It is not clear.

2. Please add brief description on CIMP-H in section “Genomic variations analysis”. What is referred by H. Biological explanation would be needed for better understanding of your research hypothesis.

3. A calculation formula or explanation need to be shown on the “Non-CIMP-H patches in MSI (76%)” in section “Genomic variations analysis”. In the previous sentence, it was described as “accounting for 59% of MSI patches”. So, I assumed 100 – 59 = 41. Your description was 76.

7. PLOS authors have the option to publish the peer review history of their article (what does this mean?). If published, this will include your full peer review and any attached files.

Reviewer #1: No

---

## [Author Response · Author response to Decision Letter 1]

30 Jun 2024

We thank the editor-in-chief, the academic editor, and the reviewer for their thorough and insightful comments. We have carefully considered each comment and have made the appropriate additions and changes in the paper to reﬂect them as described below.

Should you have any further questions, please do not hesitate to contact us. 

Best regards, 

The Authors

Reviewer 1 comments:

1. Please add a description on what is referred by the positive in Fig 4. MSI? It is not clear.

We apologize for this unclear description. In response to the reviewer's comment, we added the description of what is referred to by the positive in Fig 4. in the "genomic variations analysis"' sub-section in the Results section.

Specifically, the following paragraph was added: 

"Figure 4 presents three molecular features of our CRC patients at the patient level, plotted against the patch classification from our baseline model. In this figure, the MSI class is denoted as the positive class, and the MSS class is denoted as the negative class."

2. Please add brief description on CIMP-H in section "Genomic variations analysis''. What is referred by H. Biological explanation would be needed for better understanding of your research hypothesis.

We thank the reviewer for this comment. In response, we have added a brief description on CIMP-H in the section "Genomic variations analysis''. Specifically, the following paragraph was added: "Drawing from Liu et al. findings [17], SNP mutations are highly frequent in MSI patients due to their deficiency in the DNA mismatch-repair mechanism. However, MSI samples exhibit significant disparities in SNP density, ranging dramatically from 10 to 17,000, with a median of 1,432. Additionally, the CIMP rate, which influences gene silencing [24], is typically high in MSI patients. Specifically, 60% of MSI patches are categorized as CIMP-High (CIMP-H), while the remaining 40% are non-CIMP-H and categorized as CIMP-low and non-CIMP.''

3. A calculation formula or explanation need to be shown on the "Non-CIMP-H patches in MSI (76%)'' in section "Genomic variations analysis''. In the previous sentence, it was described as "accounting for 59% of MSI patches''. So, I assumed 100 – 59 = 41. Your description was 76.

We apologize for the unclear description. We revised the text to better clarify that these percentages are associated with different aspects and do not cover two parts of the group. Therefore no need to sum up to 100%. Specifically, CIMP-H accounts for 59\\% of the MSI patches. From the rest 41% which are MSI but not CIMP-H, a substantial portion (76%) was incorrectly classified as MSS (negative class).

We added the following paragraph to the text: "Highly methylated (CIMP-H) samples are rare in MSS, present in only 1% of the MSS patches, but are prevalent in MSI, accounting for 59% of the MSI patches. Notably, among the MSI patches that are non-CIMP-H, a substantial portion (76%) was incorrectly classified as MSS (negative class).''

---

## [Decision Letter · Decision Letter 2]

18 Jul 2024

PONE-D-24-07948R2Exploring the Interplay Between Colorectal Cancer Subtypes Genomic Variants and Cellular Morphology: A Deep-Learning ApproachPLOS ONE

Dear Dr. Freiman,

Thank you for submitting your manuscript to PLOS ONE. After careful consideration, we feel that it has merit but does not fully meet PLOS ONE’s publication criteria as it currently stands. Therefore, we invite you to submit a revised version of the manuscript that addresses the points raised during the review process.

We look forward to receiving your revised manuscript.

Kind regards,

Tao Huang

Academic Editor

PLOS ONE

Reviewers' comments:

Reviewer's Responses to Questions

**Comments to the Author**

1. If the authors have adequately addressed your comments raised in a previous round of review and you feel that this manuscript is now acceptable for publication, you may indicate that here to bypass the “Comments to the Author” section, enter your conflict of interest statement in the “Confidential to Editor” section, and submit your "Accept" recommendation.

Reviewer #1: All comments have been addressed

Reviewer #2: All comments have been addressed

2. Is the manuscript technically sound, and do the data support the conclusions?

Reviewer #1: Yes

Reviewer #2: Partly

3. Has the statistical analysis been performed appropriately and rigorously? 

Reviewer #1: Yes

Reviewer #2: Yes

4. Have the authors made all data underlying the findings in their manuscript fully available?

Reviewer #1: Yes

Reviewer #2: No

5. Is the manuscript presented in an intelligible fashion and written in standard English?

Reviewer #1: Yes

Reviewer #2: Yes

6. Review Comments to the Author

Reviewer #1: Thank you for the opportunity to review your interesting works. I would recommend your manuscript to be published in PLOS ONE.

Reviewer #2: The authors have picked a fascinating topic i.e, genomic and image data for subtype classification and to study their relationship using AI, yet the presentation of their work raises many questions.

1. Why did writers refer to their model as a 'Biologically-primed model'?

2. “consistent distribution of the genomic variations across each fold”. Are you categorizing subtypes of Colorectal Cancer or examining genetic variations?

3. The elucidation of model is presently ambiguous. It would be advantageous if the authors could furnish a flow diagram that delineates the complete process, encompassing data preprocessing to model validation.

4. What is the number of SNPs or CpG sites used for model development?

5. The authors have not provided a clear explanation of how they have combined the genomic and imaging data.

7. PLOS authors have the option to publish the peer review history of their article (what does this mean?). If published, this will include your full peer review and any attached files.

Reviewer #1: No

Reviewer #2: **Yes: **ASIM BIKAS DAS

---

## [Author Response · Author response to Decision Letter 2]

27 Jul 2024

Dear Prof. Chenette, Editor in Chief, and Prof. Huang, Academic Editor, 

PLOS ONE,

We thank the editor-in-chief, the academic editor, and the reviewer for their thorough and insightful comments. We have carefully considered each comment and have made the appropriate additions and changes in the paper to reﬂect them as described below.

Should you have any further questions, please do not hesitate to contact us. 

Best regards, 

The Authors

Reviewer 1 comments

1. "Thank you for the opportunity to review your interesting works. I would recommend your manuscript to be published in PLOS ONE.''

Thank you very much for your kind words and for taking the time to review our work. We are delighted you find our manuscript interesting and appreciate your recommendation for publication in PLOS ONE.

Reviewer 2 comments

1. "The authors have picked a fascinating topic i.e, genomic and image data for subtype classification and to study their relationship using AI.''

We thank the reviewers for the kind feedback. We are pleased to hear that you find our topic on using AI for genomic and image data in subtype classification and their relationship fascinating.

2. "Why did writers refer to their model as a 'Biologically-primed model'?''

We thank the reviewer for raising this issue. The main difference between our proposed models and those previously proposed for CRC subtype classification lies in how they handle genomic variations within each subtype. Previous models consider only the CRC subtypes as potential classes, ignoring the heterogeneity within each subtype. In contrast, our model architecture accounts for this genomic heterogeneity, allowing for a more nuanced and accurate classification. To better reflect this, we term our model ``biologically-primed,'' as it integrates biological variations within subtypes, leading to a more comprehensive and precise understanding of CRC subtypes. 

To better clarify that, we added the following explanation to our introduction: 

``We examined this hypothesis by developing and evaluating ``biologically-primed'' CNN classification models that account for the potential correlation between the genomic variations and the imaging phenotype. This is in contrast to previously proposed models for CRC subtype classification which considered only the CRC subtypes as potential classes, ignoring the heterogeneity within each subtype. To better reflect this, we term our model ``biologically-primed,'' as it integrates biological variations within subtypes, leading to a more comprehensive and precise understanding of CRC subtypes.'' 

3. ""consistent distribution of the genomic variations across each fold''. Are you categorizing subtypes of Colorectal Cancer or examining genetic variations?''

We thank the reviewer for pointing out this ambiguity. Our models aim to categorize CRC subtypes (i.e., MSI and MSS). However, recognizing the genomic heterogeneity within each subtype, the model architecture is designed to internally classify each subtype based on its genomic variations. Therefore, we ensured that the distribution of both the CRC subtypes and their internal genomic variations is consistent across the different folds. To better reflect this, we have revised the sentence mentioned by the reviewer as follows:

``consistent distribution of the CRC subtypes and their internal genomic variations across each fold''

4. "The elucidation of model is presently ambiguous. It would be advantageous if the authors could furnish a flow diagram that delineates the complete process, encompassing data preprocessing to model validation.''

We thank the reviewer for pointing this out. In response we included a new figure (Fig. 3 in the current revision) delineating the complete process, encompassing data preprocessing to model validation.

5. "What is the number of SNPs or CpG sites used for model development?''

The number of SNPs and the CIMP category are extracted from a patient's DNA sample in the TCGA. The range of SNPs among patients varied from 20 to 5000. Based on our analysis described in the Results / Genomic Variations Analysis section, we set the SNP threshold to 1200 and divided the CIMP categories into CIMP-H and non-CIMP-H. To better clarify this aspect, we added the following details to our Results / Genomic Variations Analysis section:

``The number of SNPs and the CIMP category are extracted from a patient's DNA sample in the TCGA. The range of SNPs among patients varied from 20 to 5000.'' 

6. "The authors have not provided a clear explanation of how they have combined the genomic and imaging data''

We thank the reviewer for pointing this out. As shown in Fig. 2b in the manuscript, our model does not directly use genomic data as input. Instead, we represent the MSI class as two distinct classes based on their genomic variations. The genomic variation information is utilized during model training to label the MSI patches as either MSI$_1$ or MSI$_2$. To better clarify this aspect we added the following to our introduction:

``It is important to highlight that our model does not directly use genomic data as input. Instead, we represent the CRC subtype class as two distinct classes based on their genomic variations. The genomic variation information is utilized during model training to label the MSI patches as either MSI$_1$ or MSI$_2$. Therefore, while our training phase incorporates molecular subtype information, the inference process depends exclusively on the H\\&E images, with no additional data used. ''

---

## [Decision Letter · Decision Letter 3]

12 Aug 2024

Exploring the Interplay Between Colorectal Cancer Subtypes Genomic Variants and Cellular Morphology: A Deep-Learning Approach

PONE-D-24-07948R3

Dear Dr. Freiman,

We’re pleased to inform you that your manuscript has been judged scientifically suitable for publication and will be formally accepted for publication once it meets all outstanding technical requirements.

Kind regards,

Tao Huang

Academic Editor

PLOS ONE

Additional Editor Comments (optional):

Reviewers' comments:

Reviewer's Responses to Questions

**Comments to the Author**

1. If the authors have adequately addressed your comments raised in a previous round of review and you feel that this manuscript is now acceptable for publication, you may indicate that here to bypass the “Comments to the Author” section, enter your conflict of interest statement in the “Confidential to Editor” section, and submit your "Accept" recommendation.

Reviewer #1: All comments have been addressed

Reviewer #2: All comments have been addressed

2. Is the manuscript technically sound, and do the data support the conclusions?

Reviewer #1: Yes

Reviewer #2: Yes

3. Has the statistical analysis been performed appropriately and rigorously? 

Reviewer #1: Yes

Reviewer #2: Yes

4. Have the authors made all data underlying the findings in their manuscript fully available?

Reviewer #1: Yes

Reviewer #2: Yes

5. Is the manuscript presented in an intelligible fashion and written in standard English?

Reviewer #1: Yes

Reviewer #2: Yes

6. Review Comments to the Author

Reviewer #1: (No Response)

Reviewer #2: (No Response)

7. PLOS authors have the option to publish the peer review history of their article (what does this mean?). If published, this will include your full peer review and any attached files.

Reviewer #1: No

Reviewer #2: **Yes: **ASIM BIKAS DAS

---

## [Editor Report · Acceptance letter]

30 Aug 2024

PONE-D-24-07948R3 

PLOS ONE

Dear Dr. Freiman, 

I'm pleased to inform you that your manuscript has been deemed suitable for publication in PLOS ONE. Congratulations! Your manuscript is now being handed over to our production team.

Kind regards, 

on behalf of

Dr. Tao Huang 

Academic Editor

PLOS ONE